# Pathophysiology and Clinical Management of Dyslipidemia in People Living with HIV: Sailing through Rough Seas

**DOI:** 10.3390/life14040449

**Published:** 2024-03-28

**Authors:** Eleni Papantoniou, Konstantinos Arvanitakis, Konstantinos Markakis, Stavros P. Papadakos, Olga Tsachouridou, Djordje S. Popovic, Georgios Germanidis, Theocharis Koufakis, Kalliopi Kotsa

**Affiliations:** 1First Department of Internal Medicine, AHEPA University Hospital, Aristotle University of Thessaloniki, 54636 Thessaloniki, Greece; epapanta@auth.gr (E.P.); conmark@windowslive.com (K.M.); olgatsachouridou.iasis@gmail.com (O.T.); 2Division of Gastroenterology and Hepatology, First Department of Internal Medicine, AHEPA University Hospital, Aristotle University of Thessaloniki, 54636 Thessaloniki, Greece; arvanitak@auth.gr (K.A.); geogerm@auth.gr (G.G.); 3Basic and Translational Research Unit, Special Unit for Biomedical Research and Education, School of Medicine, Faculty of Health Sciences, Aristotle University of Thessaloniki, 54636 Thessaloniki, Greece; 4First Department of Pathology, Medical School, National and Kapodistrian University of Athens, 11527 Athens, Greece; stpap@med.uoa.gr; 5Clinic for Endocrinology, Diabetes and Metabolic Disorders, Clinical Centre of Vojvodina, 21137 Novi Sad, Serbia; pitstop021@gmail.com; 6Medical Faculty, University of Novi Sad, 21000 Novi Sad, Serbia; 7Second Propedeutic Department of Internal Medicine, Hippokration General Hospital, Aristotle University of Thessaloniki, 54642 Thessaloniki, Greece; thkoyfak@auth.gr; 8Division of Endocrinology and Metabolism and Diabetes Center, First Department of Internal Medicine, Medical School, AHEPA University Hospital, Aristotle University of Thessaloniki, 1 St. Kiriakidi Street, 54636 Thessaloniki, Greece

**Keywords:** HIV, dyslipidemia, metabolic syndrome, antiretroviral therapy, switching strategy

## Abstract

Infections with human immunodeficiency virus (HIV) and acquired immune deficiency syndrome (AIDS) represent one of the greatest health burdens worldwide. The complex pathophysiological pathways that link highly active antiretroviral therapy (HAART) and HIV infection per se with dyslipidemia make the management of lipid disorders and the subsequent increase in cardiovascular risk essential for the treatment of people living with HIV (PLHIV). Amongst HAART regimens, darunavir and atazanavir, tenofovir disoproxil fumarate, nevirapine, rilpivirine, and especially integrase inhibitors have demonstrated the most favorable lipid profile, emerging as sustainable options in HAART substitution. To this day, statins remain the cornerstone pharmacotherapy for dyslipidemia in PLHIV, although important drug–drug interactions with different HAART agents should be taken into account upon treatment initiation. For those intolerant or not meeting therapeutic goals, the addition of ezetimibe, PCSK9, bempedoic acid, fibrates, or fish oils should also be considered. This review summarizes the current literature on the multifactorial etiology and intricate pathophysiology of hyperlipidemia in PLHIV, with an emphasis on the role of different HAART agents, while also providing valuable insights into potential switching strategies and therapeutic options.

## 1. Introduction

Since the first case reports in 1981, human immunodeficiency virus (HIV) infection and acquired immune deficiency syndrome (AIDS) remain among the world’s greatest pandemics, with more than 39 million people living with HIV (PLHIV) and 1.3 million newly infected in 2022 [1]. The development of highly active antiretroviral therapy (HAART) has transformed AIDS into a rather long-term chronic condition through the substantial suppression of viral load, partial restoration of the immune system, and decreased fatal HIV-related illnesses [2,3]. With nearly normal life expectancy among PLHIV who receive HAART, it is estimated that 73% of HIV-infected individuals will be over 50 years old by 2030, highlighting age-related comorbidities such as metabolic syndrome (MetS) and the consequent cardiovascular disease (CVD) as an emerging problem since obesity and aging have both seen dramatic increases in prevalence throughout society [4,5,6]. MetS, characterized by abdominal obesity, high blood pressure, increased fasting glucose, increased triglycerides (TGs), and decreased high-density lipoproteins (HDLs), is highly associated with CVD [7,8]. Its prevalence ranges between 11% and 48% among PLHIV, and it is estimated that 78% of them will develop CVD at some point in life [9,10]. Moreover, individuals with HIV receiving HAART face greater risk for major metabolic-related cardiovascular events as compared with those uninfected, experiencing earlier manifestations of heart failure, while also demonstrating nearly a two-fold increased risk for myocardial infarction and a four-fold increased risk for sudden cardiac death [11,12,13,14,15].

Dyslipidemia, a cornerstone of MetS and a well-established risk factor for CVD, is responsible for roughly 50% of all cardiovascular events among PLHIV. Lipid abnormalities that include low levels of HDL, low-density lipoprotein (LDL), total cholesterol (TC), elevated TG, and oxidized LDL (oxLDL) lead to an atherogenic profile in more than 67% of women and 81% of men with HIV infection [16,17,18,19]. In more detail, the underlying mechanisms of dyslipidemia in PLHIV involve complex pathophysiological pathways associated with, in addition to traditional risk factors, both HAART treatment and HIV infection per se. HIV-encoded proteins modify the expression of regulatory genes and the function of cell membrane proteins, resulting in the accumulation of free fatty acids (FFAs) and leading to lipotoxicity, while chronic inflammation and immune activation seen in HIV infection lead to increased levels of cytokines, decreased TG clearance, increased levels of oxLDL, and alterations in lipid particle composition [20,21,22,23,24]. Moreover, as adipose tissue serves as a reservoir for HIV, inflammation plays a key role via a continuous interplay between CD4+ T cells, macrophages, and adipocytes, further dysregulating lipid metabolism; specific HAART categories are implicated in the redistribution of adipose tissue in the form of lipodystrophy [25,26,27,28,29]. In addition to the aforementioned frequent clinical entity in PLHIV, antiretroviral treatment can even promote dyslipidemia with numerous molecular mechanisms, with some protease inhibitors (PIs), nucleoside reverse transcriptase inhibitors (NRTIs), and non-nucleoside reverse transcriptase inhibitors (NNRTIs) having the most profound effects on the lipidemic profile [30,31]. Severe drug–drug interactions between HAART and statins, combined with limited evidence of newly introduced hypolipidemic agents in HIV population, bring to the surface the importance of switching from older HAART regimens to lipid-friendly ones, highlighting that the management of dyslipidemia in PLHIV is a rather demanding issue.

The purpose of the present review is to summarize the current literature on the multifactorial etiology and intricate pathophysiology of hyperlipidemia in PLHIV, focusing on the role of different HAART agents, while also providing valuable insights into possible switching strategies and treatment options. 

## 2. The Molecular Mechanisms of HIV-Associated Dyslipidemia

### 2.1. The Role of HIV Viremia and Inflammation

HIV infection induces lipid abnormalities via several mechanisms, with the shedding of viral proteins, immunological activation, and persistent inflammation prevailing among them. Vpr, an HIV protein responsible for viral replication that plays a multifactorial role by inhibiting the peroxisome proliferator-activating receptor-γ (PPARγ) and stimulating the glucocorticoid receptor (GR) and the liver X receptor-a (LXR-α), leads to preadipocyte differentiation, dysregulation, and the overaccumulation of FFA, while it also decreases hepatic fatty acid oxidation and reduces hepatic VLDL-TG exportation, ultimately leading to lipodystrophy [32,33,34,35,36]. Tat, a regulatory HIV protein involved in viral transcription, also interferes with normal cholesterol turnover and esterification through the up-regulation of genes encoding 7-dehydrocholesterol reductase (DHCR7), resulting in increased levels of free cholesterol, TC, and cholesteryl esters [21,37,38]. Furthermore, Tat induces the expression of adhesion molecules and stimulates monocyte chemoattractant protein-1 (MCP-1)-mediated monocyte transmigration, perpetuating immune activation and inflammation [39,40]. Another protein shed by HIV, Nef, holds an important role in the viral replication and immunological escape of HIV, while also exhibiting a bifactorial role in cholesterol bioavailability. It stimulates cholesterol biosynthesis and inhibits its efflux by suppressing the activity of the ATP-binding cassette transporter protein A1 (ABCA1), while at the same time disrupting caveolin-dependent cholesterol transport in infected macrophages, resulting in decreased HDL levels and an abundance of lipid rafts. It also reduces endothelial nitric oxide (NO) production; increases inflammatory cytokine release, such as IL-6 and TNF-α and promotes the secretion of MCP-1 from endothelial cells, inducing endothelial apoptosis and promoting atherosclerotic plaque rupture and the development of acute thrombus [24,41,42,43,44,45,46]. 

Inflammation plays a crucial role in dyslipidemia and associated atherosclerosis, serving both as a cause and as a consequence. In HIV infection, the subpopulation of CD14+/CD16+ pro-inflammatory monocytes predominates, as indicated by the increased ratio of CD4+/CD8+, expressing activation markers and molecules presenting antigens such as CD38, CD69, CD11b, and CD86, resulting in tissue migration and turnover to cholesterol-overloaded dysfunctional macrophages. These so-called foam cells found in adipose tissue, together with activated CD4 and CD8 T cells and NK and NKT cells, result in the production of inflammatory mediators such chemokines CCL2, CCL5 and CX3CL1, c-reactive protein (CRP), IL-6, IL-8, IL-1β, IL-18, IL-2, IFN-γ, IL-17A, and TNF-α, altering adipose cell function, impairing reverse cholesterol transport, and reducing HDL and apolipoprotein A-I (apoA-I) particle numbers [47,48,49,50,51,52]. Furthermore, these foam cells are highly concentrated in NADPH oxidases, enzymes that, under a hyperlipidemic environment, up-regulate and form oxLDL, inducing endoplasmic reticulum (ER) stress and the production of reactive oxygen species (ROS), exacerbating both inflammation and foam cell formation, which are strong promoters of atherogenesis [53,54,55]. Along the same line, another mechanism which promotes the ongoing inflammation in HIV-associated dyslipidemia is the activation of NLRP3 inflammasome. In fact, once pathogen-associated molecular patterns (PAMPs) and danger-associated molecular patterns (DAMPs) recognize HIV particles through toll-like receptors (TLRs), the formation of an inflammasome occurs, resulting in a cascade of cytokines and the further production of IL-1β and IL-18 [56,57]. Additionally, HIV infection is characterized by the production of interferon-alpha (IFN-α) as an attempt of the immune system to prevent viral entry and inhibit viral replication. Both IFN-α and TNF-α are associated with impaired oxidation of plasma FFAs, contributing to enhanced hepatic re-esterification and elevated plasma levels of TGs. Another important branch of inflammation-induced dyslipidemia is the alteration of the gut microbiome that occurs in HIV infection. In fact, damage to the intestinal epithelium, microbial translocation, and subsequent production of microbial metabolites and toxins such as lipopolysaccharide, along with the down-regulation of normal flora bioproducts such as butyrate, contribute to persistent inflammation, as measured by circulating soluble CD14 (sCD14), soluble CD163 (sCD163), and CRP, leading to an increased TG/HDL ratio [58,59,60,61,62].

### 2.2. The Role of Antiretroviral Treatment

The introduction of HAART as a combination of three antiretroviral agents, typically including two NRTIs and one PI, NNRTI or integrase strand transfer inhibitor (INSTI), has revolutionized the treatment of HIV infection. However, nowadays, the long-term metabolic side effects of those regimens, such as dyslipidemia, have become a concern [63]. The impact of HAART on the lipid profile is often challenging to determine, considering the significant variability between different classes of ART drugs and drugs within the same class, as well as the multidrug nature of HIV treatment itself. However, altered lipid parameters in HAART-experienced patients, as expressed by high levels of TG, LDL, and apolipoprotein C-III, persisting even 3 years after the initiation of HAART, have raised questions about the underlying etiology [64,65]. Furthermore, studies from the pediatric population that demonstrated a prevalence of dyslipidemia of up to 70% after 6–150 months of treatment, as well as studies that showed a nearly 15% increase in the prevalence of dyslipidemia and a significantly higher TG/HDL ratio 6 months after the initiation of HAART, highlight the eminent need of pathophysiological interpretation, especially in individuals with multidrug-resistant HIV [66,67,68]. The main reason behind the ongoing lipid dysregulation lies in the combination of continuous inflammation and immune activation, as HAART achieves viral suppression but not elimination, mitochondrial dysfunction, and altered distribution of adipose tissue. In fact, adipose tissue and lipodystrophy syndrome, which manifests as lipohypertrophy with abdominal and dorsocervical fat accumulation or lipoatrophy with subcutaneous fat loss, hold a crucial role in dyslipidemia [69] (Figure 1).

#### 2.2.1. Protease Inhibitors

PIs are mostly associated with lipohypertrophy and have numerous effects on lipid levels, with a substantial elevation in TG and VLDL levels, especially ritonavir (RTV), lopinavir (LPV), and saquinavir (SQV); and little to no effect on LDL and HDL levels, especially between generally lipid friendly darunavir (DRV) and atazanavir [70,71]. In general, PI-based regimens have a trend to develop greater atherosclerosis compared to non-PI-based regimens, as demonstrated by the increase in the intima media thickness (IMT) and the development of atheromatous plaques. More specifically, PIs inhibit lipolysis by altering lipoprotein lipase (LPL) activity, resulting in reduced TG uptake in adipocytes and elevated plasma TG levels [72,73]. Furthermore, they inhibit the nuclear localization of sterol response element binding protein-1 (SREBP-1) in adipocytes, resulting in the down-regulation of PPARγ, impaired adipocyte differentiation, and insufficient lipid removal from circulation, while concomitantly promoting the nuclear localization of SREBP-1 in hepatocytes, resulting in excessive fatty acids synthesis [74,75]. In addition, the inhibition of proteasomal apolipoprotein B (apoB) degradation and increased ER stress have been observed in cultures of hepatocytes and rat hepatocytes, respectively, resulting in increased VLDL and lipodystrophy [76,77,78]. It is also worth mentioning that ritonavir impairs endothelial function via reduction in adipose mass, and endothelial leptin receptor-dependent increases in NADPH oxidase 1 (Nox1), C-C chemokine receptor type 5 (CCR5), and inflammation, reducing nitric oxide bioavailability, while also indicating potential avenues for limiting human immunodeficiency virus infection [79]. Along the same line, chronic exposure to HIV-derived protein Tat impairs endothelial function via the indirect alteration in fat mass and Nox1-mediated mechanisms, providing potential targets to improve vascular function in HIV infection-associated CVD [80].

In a cross-sectional study with 17,852 participants, individuals who received PI regimens were associated with higher levels of TC and TG than HAART-naive patients, while those receiving the dual-PI regimen had higher levels of TG, TC, LDL, and the TC/HDL ratio [81]. In a within-class comparison, RTV-containing regimens were associated with higher levels of TC and TG and a higher TC/HDL ratio than indinavir (IDV)-containing regimens, while nelfinavir (NFV)-containing and SQV-containing regimens were associated with a reduced risk of lower HDL levels and a lower TC/HDL ratio, respectively [82]. As implied, RTV demonstrated the most significant effect on lipid parameters, even 1 week after initiation of treatment, as indicated by the elevation of 146% TG level and the increase of 159% VLDL level, even in healthy normal individuals [83,84]. Interestingly, its lipidemic effect appeared to be dose dependent, given the fact that, combined with other PIs, a low-booster dose resulted in increased TG and LDL levels by 26% and 16%, respectively, while in full-dose, TGs increased by 83%, FFAs by 30%, and VLDL by 33% [85,86]. In RTV-boosted LPV (LPV/r) regimens, a further 28%-to-108% increase in fasting and non-fasting TG levels was observed, which, combined with an increase of 25% in LDL levels and a concomitant reduced size of LDL particles, could be associated with high atherogenicity [87,88]. Although some studies demonstrated pretreatment baseline lipid values and LPV plasma concentration levels as important risk factors for induced hyperlipidemia, other studies did not verify this observation [89,90,91]. It is also worth mentioning that Amprenavir (APV) and NFV increased TG and LDL levels to a lesser extent than RTV or LPV/r in HIV-infected patients [92,93]. Moreover, ATV is considered to be among the most lipid-friendly PIs and has shown favorable lipid outcomes. In fact, some studies demonstrated the absence of a harmful effect on TG levels after 48 weeks of administration, even at a dose of up to 500 mg, while some others showed a significant decrease of 46% in TG levels, with a concomitant improvement of 18% in TC levels, during the first 24 weeks after switching to the atazanavir-based regimen [94,95]. Newer PIs such as DRV share the beneficial effect of ATV on hyperlipidemia. Data from the ARTEMIS study that included 689 patients showed that DRV had smaller median increases in TG and TC levels, +1.8 mg/dL and +10.8 mg/dL, respectively, compared to LPV, 10.8 and 16.2 mg/dL, respectively; however, in POWER studies enrolling heavily pretreated patients, a 15% increase in TG levels was observed [96,97]. Along the same line, in a phase 4 randomized exploratory study, DRV demonstrated an increase in apoA-I and, consequently, favorable changes in HDL levels, especially in HIV individuals with a low CD4+ cell count, compared to ATV [98].

#### 2.2.2. Nucleoside Reverse Transcriptase Inhibitors

NRTIs, especially the thymidine analogues stavudine (d4T) and zidovudine (ZDV), have been implicated with dyslipidemia, mainly with adipose tissue alteration and induced lipoatrophy [99]. The main pathophysiological mechanism involved is mitochondrial dysfunction and cell toxicity through the inhibition of DNA polymerase gamma and oxidative phosphorylation, increasing mitochondrial ROS production [100,101]. The increased incorporation and ineffective exonuclease removal of highly toxic dideoxy NRTI compounds presumably constitute another important branch of mitochondrial dysregulation, while insufficient respiratory chain activity and ATP synthesis, as an index of mitochondrial dysfunction, have been reported in lipodystrophic patients receiving NRTIs [102,103,104]. ZDV has especially been associated with the inhibition of mitochondrial adenylate kinase, adenosine nucleotide translocator, and electron transport chain, promoting ROS production [105,106]. Highlighting the great influence of NRTIs in adipose tissue, the expression levels of significant adipogenic factors such as PPAR-γ, SREBP-1, CCAAT/enhancer-binding protein alpha (C/EBP-α), adiponectin, and leptin were abnormally low, while cytokine levels of IL-6 and TNF-α, being produced by stressed adipocytes and immune cells, were notably high. Furthermore, NRTIs were associated with depleted cellular mitochondrial DNA (mtDNA) and mitochondrial proliferation in adipocytes, as quantified by cellular mtDNA copy number and mitochondrial mass measurements, findings highly suggestive of lipoatrophy [107]. Similar findings with a nearly 68% reduction in mtDNA/nuclear DNA levels were observed in peripheral blood mononucleated cells of HIV individuals treated with NRTIs, while the significant inhibition of mitochondrial gene expression was demonstrated after a 2-week NRTI-based regimen even in HIV-negative patients [107,108]. Interestingly, some studies link these effects with lamivudine (3TC) to a lesser extent compared to didanosine and d4T [109].

A prospective multicenter study that enrolled 873 HIV individuals who switched from d4T to tenofovir (TDF) demonstrated a sustained reduction in the median levels of TC (−17.5 mg/dL), LDL (−8.1 mg/dL), and TG (−35 mg/dL), with the greatest reduction observed among those with higher baseline values [110]. In fact, even a lower dose of d4T (30 mg b.i.d. instead of 40 mg b.i.d.) showed a clinically significant improvement of lipid parameters in HIV-treated patients [111]. Similar findings have been recorded with ZDV, which was associated with higher levels of TC and LDL compared to other first-line agents in China, suggesting a preemptive switch from thymidine analogues to other regimens to prevent further progression of dyslipidemia and lipoatrophy [112,113]. Furthermore, Abacavir (ABC) could be an alternative option of d4T and ZDV, as it has shown a positive effect in increasing limp fat and partially resolving lipoatrophy, even though an unfavorable lipid outcome has been observed, with higher levels of TG (25 mg/dL versus 3 mg/dL) and TC (34 mg/dL versus 26 mg/dL), as compared with TDF at 48 weeks, according to the results from the ACTG 5202 study [114,115]. However, the increased overall cardiovascular risk associated with ABC limits the benefits of potential switching to ABC-based regimens [116]. 3TC, TDF, and the newer agent tenofovir alafenamide (TAF) are the NRTI representatives exhibiting the most lipid-friendly profile, outmatching the rest of NRTIs in all lipid outcomes, such as TG, TC, LDL, and HDL levels [71]. Notably, a recent prospective cohort study of 1446 HIV individuals who switched from TDF to TAF had a mean weight increase of +0.5 kg at 144 weeks and a significant increase in TC (+7.9 mg/dL) and TG (+11.2 mg/dL), with no differences in the TC/HDL ratio [117]. In addition, another recent observational, single-center study with 61 HIV individuals who switched from TDF to TAF demonstrated a significant increase in TC 178 ± 38 to 194 ± 40 mg/dL, LDL levels 117 ± 32 to 137 ± 36 mg/dL, and average weight, as well as an increase in HDL levels 45 ± 12 to 48 ± 13 mg/dL, indicating that, despite the overall superiority of TAF in terms of stability and bioavailability, a personalized therapeutic approach regarding the metabolic risk should be taken into account [118].

#### 2.2.3. Non-Nucleoside Reverse Transcriptase Inhibitors

NNRTIs have also shown an effect in dyslipidemia with increased TC, LDL, and TG levels and increased HDL levels, thus counterbalancing the overall lipid risk profile due to mitochondrial dysfunction, as established by detecting increased mitochondrial mass and decreased mitochondrial membrane potential [119,120,121]. Efavirenz (EFV) has especially been associated with increased ROS production and reduced ATP synthesis through inhibition of complex I, combined with induced hepatic cell apoptosis through modified cytochrome c and caspase 9 activity [104,122]. Furthermore, EFV can serve as a potent pregnane X receptor (PXR) selective agonist, inducing target gene expression, for instance, for the fatty acid transporter CD36 gene, resulting in increased lipid uptake and cholesterol biosynthesis in cells [123]. Indeed, the unfavorable lipid effect of NNRTIs, especially EFV, has been demonstrated in a 6-year prospective observational study of 433 immunosuppressed HIV individuals, which demonstrated high TC and TG levels, as well as increased TC and LDL levels as compared with ATV/ritonavir (ATV/r) treatment [124,125]. On the contrary, some other studies demonstrated favorable HDL and apoA-I levels of EFV as compared with ATV/r, as well as the superiority of EFV as compared with LPV in terms of TG levels [126,127]. Additionally, in a between-class comparison, the 2NN study evaluated the different lipid effects of two NNRTIs, EFV and nevirapine (NVP), in combination with two NRTI (d4T and 3TC), and showed a greater increase in TG, TC, and LDL levels in the EFV arm, as compared with the NVP arm. Similar data have been extracted from the SCOLTA study enrolling 490 HIV individuals, within which switching from EFV to rilpivirine (RPV), another NNRTI, demonstrated a statistically significant improvement in TC, TG, and LDL levels and increased the TC/HDL ratio in 12 months [128]. Finally, an interesting study of 50 HIV individuals who switched from a lipid-friendly NNRTI (NVP) to another lipid-friendly NNRTI (RPV) showed a significant reduction at week 24 in the mean TC (−12 mg/dL), LDL (−6.5 mg/dL), and HDL levels (−5 mg/dL), with TG levels remaining rather stable, highlighting the challenging nature of HAART switching [129].

#### 2.2.4. Integrase Inhibitors

INSTIs seem to exert minimal or negligible influence on lipid levels, even after long-term use, highlighting the beneficial role of these agents in dyslipidemia after switching from other HAART regimens, as suggested by current guidelines [130]. A recent meta-analysis of randomized controlled trials comparing integrase inhibitors with other antiretroviral classes (EFV-based or PI-based therapies) in naive HIV patients, demonstrated that INSTIs led to decreased TC (MD −13.44 mg/dL), LDL (MD −1.37 mg/dL), HDL (MD −5.03 mg/dL), and TG levels (MD −20.70 mg/dL). However, a well-established risk of considerable weight gain among HIV individuals has been associated with INSTI-based treatment, compared to PIs and NNRTIs [131,132,133]. The pathophysiological pathway behind the aforementioned outcome is yet to be established; however, a low CD4 count, high viral load, and substantial weight loss before the initiation of HAART were associated with greater weight gain, implying that superior immune reconstitution in individuals with more advanced HIV infection appears to be an independent risk factor for INSTI-induced fat accumulation [134,135]. Recent studies have demonstrated that dolutegravir (DTG) and, to a lesser extent, Raltegravir (RAL) are associated with activation of lipogenic and adipogenic pathways, increased lipid accumulation, induced mitochondrial dysfunction and oxidative stress, low leptin and adiponectin secretion, and elevated peri-adipocyte fibrosis [136,137]. However, these adipose tissue alterations do not reflect unfavorable lipid outcomes, unlike insulin resistance [138]. Indeed, in the ACTG A5260s study, ART-naive patients undergoing RAL treatment presented with a rapid two-fold increase in insulin resistance, similar to that observed with ATV/r and DRV/r, but on the contrary, another prospective randomized study highlighted the superiority of RAL in all fasting lipid measurements, including TC, TG, non-HDL, and LDL, as compared with the two ritonavir-boosted PIs [139,140]. Furthermore, a Greek cohort study by Pantazis et al. demonstrated that INSTIs, especially DTG and RAL, as compared with elvitegravir (EVG), led to faster and more profound weight gain in comparison with PIs and NNRTIs, with a mean expected weight gain of 6kg in the INSTI-based regimen group, while a cohort study of the RESPOND study group, with 4577 HIV individuals, demonstrated that elvitegravir/cobicistat (ELG/c) and RAL were associated with a higher incidence of dyslipidemia, as compared with DTG [141,142]. Apart from DTG, second-generation INSTIs such as bictegravir (BIC) share the same lipid-friendly profile, although significant weight gain has been recorded, with a study comparing DTG + 3TC to BIC/FTC/TAF, demonstrating a significant decrease in TG levels (MC −14 mg/dL) and increased HDL levels (MC + 3 mg/dL) in the DTG group, with a significant decrease in LDL levels (−13 mg/dL) in the BIC group [143]. Finally, another second-generation INSTI, cabotegravir (CAB), in combination with RPV, had a promising lipid effect with a significant increase in HDL levels and a decrease in the TC/HDL ratio, but little to no effect in LDL levels, regardless of the regimen prior to switching [144] (Table 1).

## 3. Treatment of Dyslipidemia

The treatment of dyslipidemia in PLHIV reflects the eminent need to address its most common clinical consequence, atherosclerotic cardiovascular disease (ASCVD). ASCVD risk stratification in PLHIV is usually performed by assessing risk scores from the general population, such as the Framingham Heart Study (FHS-CVD), the Pooled Cohort equations of the American College of Cardiology/American Heart Association (PCE), and the Systematic Coronary Risk Evaluation High-Risk Equation (SCORE), in an attempt to detect those with a high or very high risk of ASCVD early on [145]. Although practical and essential, the aforementioned scores systematically underestimate the CVD risk of PLHIV, especially among low/moderate risk groups, leading to inadequate or delayed treatment initiation [146]. Furthermore, although the therapeutic approach to dyslipidemia in HIV individuals aligns with that of the general population, in PLHIV, potential interactions between lipid-lowering drugs and antiretroviral treatment should be taken into account. The initial step of lipid management involves endorsing lifestyle modifications, followed by the introduction of lipid-lowering therapy; however, in individuals with an increased risk of cardiovascular disease, switching from HAART to more lipid-friendly regimens has a pivotal role (Table 2).

### 3.1. Statins

Statins are the most commonly prescribed lipid-lowering agents and are considered the first drug of choice in PLHIV to reduce the risk of ASCVD, with seven statins currently available on the market, divided into generations depending on their origin, their synthetic compounds, and their hydrophilic or lipophilic properties [154]. Statins possess pleiotropic properties in addition to LDL reduction, consisting of inflammation deterioration, immune activation, oxidative stress, and endothelial dysfunction—a game-changing ability given the presence of persistent inflammation in HIV infection [155]. Notably, the Johns Hopkins HIV clinical cohort that enrolled 1538 virally suppressed HIV individuals under statin treatment demonstrated a reduced risk of all-cause mortality after adjusting for CD4 count, HIV-1 RNA, hemoglobin, and cholesterol levels at the start of HAART, age, race, HIV risk group, prior use of ART, year of HAART start, NNRTI versus PI-based ART, prior AIDS-defining illness, and viral hepatitis coinfection [156]. The aforementioned data are strengthened by a recent meta-analysis of 36,253 HIV individuals undergoing statin treatment, where statin use was independently statistically correlated with a reduced mortality risk in PLWH [147]. 

#### 3.1.1. The Role of Statins in Inflammation

Numerous pathophysiological pathways are involved in the pleiotropic effect of statins. As statins competitively inhibit 3-hydroxy-3-methylglutaryl-coenzyme A (HMG-CoA), which is responsible for the end-stage production of mevalonate, a proposed pathophysiological pathway involves the diminished formation of important isoprenoid intermediates such as farnesyl pyrophosphate (FPP) and geranylgeranyl pyrophosphate [157]. These molecules are responsible for prenylation, a process that affects numerous signal transduction molecules in vascular and myocardial signaling pathways, such as small guanine triphosphate (GTP)-binding proteins, which regulate pro-atherogenic pathways and the expression of pro-inflammatory cytokines; and directly activate PPAR-γ in platelets, inflammatory cells, vascular wall cells, and cardiomyocytes [158,159]. Furthermore, statins up-regulate endothelial nitric oxide synthase [82], inducing enhanced NO bioavailability and promoting its vasodilatory, anti-inflammatory, and anti-atherogenic effects. In fact, the inhibition of Rho kinases geranylgeranyl phosphorylation and activation of the PI3-Akt protein kinase pathway are both associated with the increased expression of the *eNOS* gene in human endothelial cells, while the polyadenylation of eNOS mRNA and down-regulation of caveolin-1 expression led to stabilized eNOS mRNA and prolonged activation of eNOS [159,160]. In vitro studies have also demonstrated the significant effect of statins on inflammatory cells per se, interfering with the interaction between vascular smooth muscle cells (VSMCs) and monocytes, resulting in a decreased synergistic production of pro-inflammatory cytokines, particularly IL-6 [161,162]. Interestingly, lovastatin appears to down-regulate nuclear factor kappa B (NF-κB) and activator protein-1 (AP-1) in a dose-dependent manner, with concomitant suppression of key chemokines, including those regulated upon activation normal T-cell expressed and secreted (RANTES) and the MCP-1, resulting in the reduced production of IL-2, IL-4, and IFN-γ, and ultimately decreasing the inflammatory cell infiltration of arterial walls. Furthermore, statins reduce oxLDL and increase apoA-I levels, decreasing the expression of E-selectin, intracellular adhesion molecule-1 (ICAM-1), and vascular cell adhesion molecule (VCAM), eventually reducing TNF-α, IL-6, and CRP levels [163,164,165]. Several studies have also evaluated the circulating biomarkers associated with advanced atherosclerosis in people with HIV, with numerous statins having inconsistent reduction patterns of different biomarkers [166]. In a study of 98 individuals with HIV virologically suppressed, atorvastatin 20 mg daily reduced oxLDL by 33%, sCD14, sCD163, CRP, and markers of T-cell and monocyte activation [167], while a study by Calza et al. showed a substantial reduction of CRP, IL-6, and TNF-α levels after a 12-month follow-up with 10 mg of rosuvastatin per day [167,168]. The JUPITER trial also brought to light important data regarding individuals with non-elevated LDL (LDL < 130 mg/dL), but with moderate inflammation (CRP ≥ 2 mg/L), showcasing a relative reduction of 44% in the levels of LDL and CRP compared to placebo, while the SATURN-HIV study demonstrated favorable outcomes in CD4+ and CD8+ activation markers under a 48-week rosuvastatin treatment, with a concomitant 13.2% reduction of sCD14, an event associated with a 21% decreased risk of all-cause mortality in PLHIV [148,169]. 

#### 3.1.2. The Role of Statins in Lipid Management

Although statins exhibit major pleotropic anti-inflammatory properties, their most profound effect in reducing the risk of ASCVD lies within their hypolipidemic effect. It is reported that a decrease in LDL of 2 mg/dL is associated with an average reduction of 1% in the risk of clinical events in the general population; however, some differences have been detected in PLHIV, as a reduction of 3–16% in total cholesterol was observed in PLHIV compared to non-HIV [12,170]. Furthermore, the efficacy of statin in HIV-induced dyslipidemia is well established. Calza et al. provided evidence regarding different statin options, enrolling 94 HIV individuals in PI-based treatment with hypercholesterolemia (TC > 250 mg/dL) for a duration of at least 3 months. In more detail, participants were randomized to hypolipidemic treatment with rosuvastatin 10 mg daily, pravastatin 20 mg daily, or atorvastatin 10 mg daily, with the results demonstrating a significantly higher mean decrease in TC levels with rosuvastatin (25.2%), rather than with pravastatin (17.6%) or atorvastatin (19.8%), after one year of follow-up [171,172,173,174]. Another randomized control trial assessing statin efficacy, the INTREPID study, compared pitavastatin to pravastatin and demonstrated a significantly higher reduction in LDL levels in PLHIV under 4 mg of pitavastatin versus 40 mg of pravastatin (31% and 21%, respectively) at 12 weeks of therapy, with the benefit being sustained at week 52 [175]. Most notably, reductions in TC, non-HDL, apoB, the apoB/apoA-I ratio, and the TC/HDL ratio were also significantly in favor of pitavastatin, while no differences in apoA-I, TG, or HDL were demonstrated at either week 12 or week 52. Furthermore, a recent meta-analysis demonstrated that 10 mg of rosuvastatin per day and 10 mg of atorvastatin per day provided the largest reduction in TC levels, while atorvastatin at 80 mg and simvastatin at 20 mg provided the greatest reduction in LDL levels, atorvastatin 80 mg and simvastatin 20 mg showed the greatest reduction in TG levels, and pravastatin 10–20 mg and atorvastatin 10 mg showed the largest increase in HDL levels [12]. The hallmark of statin treatment in the prevention of ASCVD risk in PLHIV, and perhaps one of the most anticipated trials in the field of dyslipidemia, was the REPRIEVE trial. REPRIEVE was a large, randomized, blinded study of pitavastatin versus placebo in more than 7500 HIV individuals, and it demonstrated, for the first time, a reduction in clinical endpoints (MACEs), defined as a composite of cardiovascular death, myocardial infarction, hospitalization for unstable angina, stroke, transient ischemic attack, peripheral arterial ischemia, revascularization, or death from an undetermined cause. The study showed an incidence of 4.81 per 1000 person-years in the pitavastatin group and 7.32 per 1000 person-years in the placebo group in MACE; it also used a pooled cohort equation for ASCVD risk stratification, thus providing an opportunity to evaluate statin benefits in those with higher and lower ASCVD risk [149].

#### 3.1.3. Drug Interactions between HAART and Statins

One of the main concerns with the use of statins in PLHIV is the potential drug–drug interactions with HAART agents, resulting in increased statin exposure and potential side effects, or decreased exposure and therapeutic failure, as well as alterations in antiretroviral bioavailability. Many HAART agents, especially PIs, pharmacokinetic boosters, and NNRTIs, share common metabolic and deactivation pathways with statins, with both drug categories serving as substrates or inhibitors of cytochrome P450, particularly CYP3A4 and CYP2C9, and organic anion-transporting polypeptides (OATPs), while complex biliary excretion and active tubular secretion through CYP3A4, OATP1B1, P-glycoprotein (P-gp), and r breast cancer resistance protein (BCRP) enhance the intricacy of statin choice.

Rosuvastatin and fluvastatin are metabolized primarily by CYP2C9, while pravastatin is minimally metabolized by P450 enzymes; therefore, they are considered safe options when combined with PIs and NNRTI, although rosuvastatin and pravastatin may demonstrate minor interactions due to the inhibition of OATP1B1 [176]. In fact, the co-administration of rosuvastatin with ATV/r has been associated with a significant increase in ATV concentration above the therapeutic threshold, with a concomitant risk of adverse drug reactions, while other studies demonstrated an increased maximum concentration of rosuvastatin when co-administered with ATV, LPV, or DRV/r of 600%, 366%, and 139%, respectively [177,178]. Therefore, if rosuvastatin is concomitantly administered with PIs, a low dose of 5 mg per day is recommended, with slow titration and close monitoring. Along the same line, studies regarding pravastatin have demonstrated an exposure decrease of 50% in patients receiving SQV/r and 40% in those receiving EFV, as well as an increase in the AUC of 33% and 81% in PLHIV under LPV/r and DRV/r, respectively; thus, guidelines recommend a suitable dose adjustment to achieve the expected benefit, especially with DRV/r [179]. Furthermore, clinically relevant interactions between fluvastatin and HAART have not been extensively documented, although co-administration with NFV and EFV could result in low plasma statin concentrations; hence, a higher initial dose is suggested [180]. Lovastatin and simvastatin have extensive metabolism first pass by CYP3A4 and are contraindicated in HIV individuals under HHART, mainly due to the severe and fatal adverse effects upon concomitant use with PI-based regimens, alongside the easy access to safer statin options [181,182].

Atorvastatin follows the same metabolic pathway, and, to a lesser extent, it serves as a substrate for OATP1B1 and shows affinity for CYP3A4 and P-gp; therefore, its concentration can differ with the co-administration of PIs or NNRTIs. In fact, different studies presented contradictory evidence due to the different pharmacokinetic interactions of atorvastatin and certain HAART agents [183]. In more detail, evidence has demonstrated an increased exposure of 79% in HIV individuals receiving SQV/r and nearly 488% with LPV/r, and thus guidelines recommend a submaximal initial dose of atorvastatin, 10 mg in ATV/r-containing regimens, 20 mg in LPV/r-containing regimens, and 40 mg DRV/r-containing regimens [176,181]. However, the potential reduction in atorvastatin AUC by 32% and 43% with ETV and EVG, respectively, highlights the necessity for increased dosage of atorvastatin, with a threshold of 80 mg per day [184]. Significant interactions due to the inhibition of CYP3A4, P-gp, and BCRP could present with the concomitant use of cobicistat to enhance ATV, DRV, and ELV, and despite data scarcity, the tendency is to initiate the lowest recommended dose, titrate carefully, and monitor for adverse effects, especially with atorvastatin and rosuvastatin [185]. Pitavastatin is mainly metabolized via glucuronidation and minimally by CYP450 enzymes, and, thus, the potential for drug interactions through the CYP450 system is reduced, rendering pitavastatin a rather safe option with no interactions expected. Similarly, INSTIs do not pose a risk for drug–drug interaction, as they exhibit weak-to-no inhibition of BCRP and other metabolizing enzymes (Table 3).

### 3.2. Ezetimibe

Ezetimibe inhibits the Niemann–Pick C1-like cholesterol transport protein (NPC1L1) at the brush border of the small intestine, leading to the up-regulation of LDL receptors and its circulatory clearance [186]. As ezetimibe does not interact with CYP3A4 and therefore is not associated with drug–drug interactions with HAART, it should be considered as a treatment option in PLHIV with statin intolerance, or as an additional therapy in those who already receive the maximum indicated statin dose and do not reach LDL therapeutic targets. Furthermore, the results of a recent meta-analysis of 13 randomized controlled trials and single-arm trials comparing rosuvastatin plus ezetimibe versus rosuvastatin monotherapy showed significant reductions in LDL (−23.89 mg/dL), TC (−26.17 mg/dL), and TG levels (−18.57 mg/dL), but no reduction in HDL levels with ezetimibe [187]. Meanwhile, other studies showed a mean decrease of −18.18 mg/dL versus −9 mg/dL in TC levels, a mean decrease of −11.16 mg/dL versus −3 mg/dL in TG levels, and a mean decrease of −17.46 mg/dL versus −9.5 mg/dL in non-HDL levels [150]. Notably, numerous studies also demonstrated favorable outcomes with ezetimibe initiation with regard to inflammation markers, particularly CRP, IL-1β and IL-18. In fact, a reduction in adipocyte size, the accumulation of pro-inflammatory cytokines, the expression of TNF-α, and the suppression of NF-kB activation have been associated with its use, making ezetimibe a rather strategic asset in the management of dyslipidemia in PLHIV [188,189,190,191].

### 3.3. PCSK9 Inhibitors

It should be mentioned that proprotein convertase subtilisin/kexin type 9 (PCSK9) inhibitors, a heterogeneous group of molecules consisting of two monoclonal antibodies, alirocumab and evolocumab; and a synthetic small interfering RNA (siRNA), inclisiran, have currently emerged as treatment options for hyperlipidemia, and they have been associated with a reduction in serum LDL levels through the up-regulation of LDL receptors and increased LDL clearance [192]. PCSK9i is used as a third-step approach in HIV individuals with an increased risk of ASCVD for those who face intolerance or severe drug–drug interactions with statins and also fall behind the recommended LDL levels with statin plus ezetimibe interventions, offering an additional 43%-to-64% reduction in LDL, alongside statins [193]. Although recent EACS guidelines included both evolocumab and alirocumab, with their safety and efficacy already having been extensively documented in the FOURIER and ODYSSEY studies for the general population, with no drug interactions having been reported, data are only available for evolocumab in PLHIV [194,195]. Among the first reports, a small single-center study with 19 HIV individuals demonstrated the favorable effects of evolocumab in LDL levels and coronary endothelial function, as measured by cine 3T MRI, setting the boundaries for the BEIJERINCK study [196]. In this study, Boccara et al. enrolled 467 PLHIV with a well-treated HIV infection and mean LDL levels at baseline 133 ± 40 mg/dL. In more detail, among the participants, 31% of them were under statins, 21% had a history of statin intolerance, and 40% had potential drug–drug interactions between statins and HAART. The results showed a persistent reduction in LDL levels of 58% through 52 weeks of exposure, with a concomitant sustained improvement in TC, TG, non-HDL, VLDL, apoB, Lp(a), and HDL levels, thus highlighting the beneficial role of PCSK9 in PLHIV with advanced hyperlipidemia [151]. Although the underlying mechanisms are still poorly understood, experimental models and clinical trials of individuals from the general population with familial hypercholesterolemia have shown a reduced expression of ICAM-1 and CCR2 in monocytes, as well as the down-regulation of TNF-α, IL-1, and IL-6 and the up-regulation of IL-10 with PCSK9i, highlighting the potential anti-inflammatory properties of PCSK9i in PLHIV [197,198].

### 3.4. Bempedoic Acid

Bempedoic acid, a recently approved hypolipidemic agent for adults with heterozygous familial hypercholesterolemia or established risk of ASCVD, has shown impressive LDL-lowering properties, either as monotherapy or in combination with ezetimibe, through the inhibition of ATP citrate lyase (ACLY) alongside direct activation of AMP-activated protein kinase (AMPK) [199,200]. Although some studies have shown favorable results for statin-intolerant patients or those under a maximally tolerated statin dose of the general population, data from HIV individuals are scarce at the present. The randomized CLEAR trial showed a reduction in LDL levels of 17.8% and 24.5% among statin-treated and statin-intolerant patients, respectively, with a concomitant reduction of 18.1% in CRP levels, while recently published data from the same trial with 13,970 statin-intolerant participants reported a reduction in all major adverse cardiovascular events with the use of bempedoic acid [152,201]. Furthermore, Ballantyne et al. enrolled 301 patients with an increased risk of ASCVD undergoing statin therapy in a phase 3, double-blind clinical trial and randomly assigned them to a fixed dose of bempedoic acid plus ezetimibe, bempedoic acid as monotherapy, ezetimibe alone, or placebo, showcasing a 36.2% reduction in LDL levels with the combination treatment, 23.2% reduction with ezetimibe, and 17.2% reduction with bempedoic acid as monotherapy [202]. The aforementioned evidence, alongside the lack of drug interactions or considerable muscle-related side effects, has led the updated EACS guidelines to enlist 180 mg of bempedoic acid once daily as a potential therapeutic approach in HIV individuals with unmet LDL goals [203].

### 3.5. Fibrates

A significant metabolic disorder and a predominant abnormal lipid characteristic in HIV infection are considered hypertriglyceridemia. The cornerstone of its treatment remains to this day lifestyle modifications, aiming at TG levels < 150 mg/dL, followed by statin treatment for HIV individuals with an increased risk of ASCVD and TG > 200 mg/dL; however, treatment for markedly elevated TG levels > 500 mg/dL, or even above this range, usually requires fibrate initiation due to the increased risk of pancreatitis [204]. Fibrates act by binding and activating the nuclear hormone receptor peroxisome proliferator-activated receptor (PPAR-α), inducing PPAR-dependent gene transcription and up-regulating lipoprotein lipase, thus limiting substrate availability in the liver for TG synthesis and increasing TG clearance [205]. On the same side, the activation of PPAR-a leads to the decreased production of pro-inflammatory mediators such as TNF-α, IL-1, IL-6, and IL-8, while also promoting the production of anti-inflammatory agents, such as IL-10, thus contributing to inflammatory retention in HIV infection [206]. Fenofibrate lacks significant interactions with ART, while gemfibrozil’s inhibition of OATP might result in increased systemic exposure when co-administered with specific HAART regimens such as LPV/r, but it can also lead to decreased hypolipidemic efficacy and increased serum concentrations of statins [207,208]. Furthermore, a study by Silverberg et al., with 6941 HIV individuals, demonstrated a substantially lower reduction in TG levels in PLHIV as compared with the general population, with great variation among individual HAART classes, −44.0% in patients receiving PI monotherapy, −26.4% in patients receiving PIs and NNRTIs, and −60.3% in patients receiving NNRTIs [209]. Amongst fibrates, gemfibrozil seems to be more effective in PWH as compared with fenofibrate, with a mean TG reduction of 80 mg/dL and 49 mg/dL, respectively; meanwhile, in combination with ezetimibe, it achieved a remarkable reduction in TG levels (from 265 ± 118 mg/dL to 149 ± 37 mg/dL) and a considerable augmentation of HDL levels (44 ± 10 to 53 ± 12 mg/dL) in comparison with statins [153,210]. Although their hypolipidemic effect is solid, fibrates failed to reduce the incidence of cardiovascular events of more than 10,000 patients with increased ASCVD risk in the PROMINENT study that was conducted with participants from the general population; therefore, similar long-scale studies in the HIV population seem to be imperative [211]. 

### 3.6. Fish Oils

Fish oils are long-chain omega-3 polyunsaturated fatty acids (PUFAs), and their two purified forms of ethyl esterized n-3 fatty acids, eicosapentaenoic acid (EPA) and docosahexaenoic acid (DHA), are commonly used to decrease TG levels and the risk of ASCVD in the general population. Their hypolipidemic effect, along with their anti-inflammatory properties that have been attributed to the inhibition of VLDL production, has been well documented in the REDUCE-IT trial. This study, in which 8179 statin-treated patients with elevated TG levels and cardiovascular disease or diabetes were randomized to 4 g of icosapent ethyl per day or placebo, showed a 25% relative and 4.8% absolute reduction in the primary end points of MACE, except death from any cause, while the REDUCE-IT biomarker sub-study demonstrated a significant reduction in serum levels of IL-1, IL-6, CRP, oxLDL, homocysteine, Lp(a), and lipoprotein-associated phospholipase A2, thus establishing EPA as a promising therapeutic option to combat hypertriglyceridemia [212,213]. Although limited, evidence for PLHIV has validated the beneficial effect of PUFAs, especially for EPA. A double-blind, placebo-controlled study that randomized 48 PLHIV under fibrate or niacin with 4 gr PUFA daily versus placebo for 12 weeks showed a reduction of 31.5 mg/dL and 7.4 mg/dL in TG levels, respectively, while the ACTG A5186 study that randomized 100 PLHIV with TG levels >400 mg/dL to 3 gr of fish oil twice daily or 160 mg of fenofibrate daily for 8 weeks demonstrated a 46% and 58% reduction in TG levels in each arm, respectively, with the combination treatment of fish oil and fenofibrate resulting in a total 65.5% reduction, achieving TG levels < 200 mg/dL in 22.7% of the patients [214,215]. Along the same line, similar evidence was also reported from a recent meta-analysis of clinical trials assessing the efficacy of PUFAs and especially EPA in PLHIV, with a 10.5 mg/dL reduction in TG levels and 11 mg/dL increase in HDL levels [216]. Therefore, fish oils seem to be a valuable asset against hypertriglyceridemia without major adverse effects or interactions; however, the pill burden in a population under polypharmacy might interfere with patient compliance. 

## 4. Conclusions

To the present day, dyslipidemia in PLHIV remains a challenge given the high prevalence of age-related metabolic diseases and the multifactorial origin of these comorbidities in inflammation and immune activation, with HIV viremia and antiretroviral treatment representing two sides of the same coin. Current antiretroviral regimens stand as safe, effective, and well-tolerated options in terms of HIV suppression; however, different HAART classes and drugs within the same class are held responsible for inducing lipid abnormalities, while considerable drug–drug interactions with major hypolipidemic agents raise concerns when treating dyslipidemia. A holistic approach of hyperlipidemia and consequent cardiovascular risk seems imperative, with the incorporation of healthy lifestyle modifications such as dietary interventions and exercise implementation demonstrating substantial but insufficient benefit. Therefore, pharmacological treatment appears to be essential, with HAART modification being the primary and crucial step, followed by hypolipidemic agent initiation. The aim of the careful evaluation of potential resistance, tolerability, adherence, or substitution of HAART seems to be a game-changing strategy. Switching from RTV-based or RTV-boosted regimens to DRV- or ATV-based ones if PIs are required, avoiding thymidine analogs, and favoring ABC and TDF in terms of NNRTI utilization, combined with shifting from EFV to NVP or RPV and incorporating lipid-friendly INSTIs, could maintain optimal viral response without the burden of associated dyslipidemia. In general, hypolipidemic interventions in PLHIV follow the guidelines for the HIV-negative population and are based on statin implementations; however, drug interactions mainly due to the CYP450 metabolism of both statins and HAART could interfere with favorable outcomes. Add-on therapy with ezetimibe, PCSK9, bempedoic acid, fibrates, or fish oils is recommended for individuals presenting with intolerance or unmet therapeutic targets, although large-scale clinical data are still scarce. An in-depth understanding of the underlying molecular mechanisms involved in HIV-associated dyslipidemia is imperative to achieve effective and personalized treatment with respect to HAART switching and hyperlipidemia, while future large-scale studies in HIV individuals that implement new lipid-lowering drugs are expected to optimize the management of metabolic-related comorbidities in PLHIV.

## Figures and Tables

**Figure 1 life-14-00449-f001:**
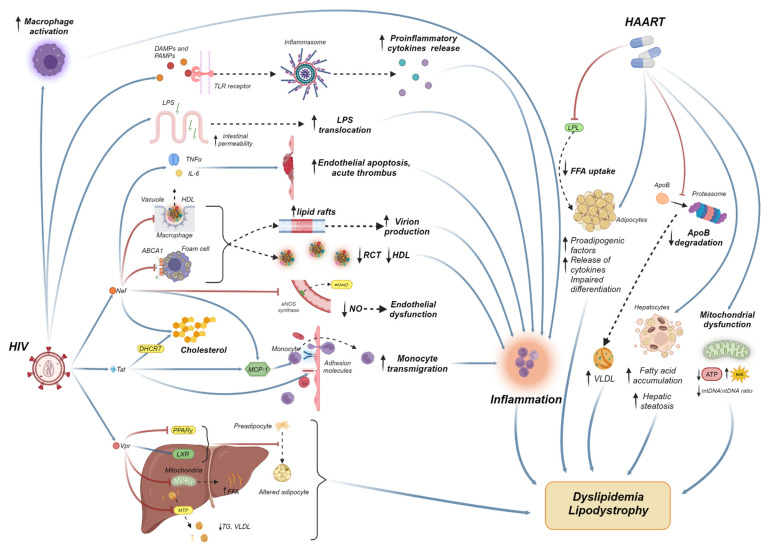
The pathogenesis of dyslipidemia in PLHIV under HAART. HIV induces macrophage activation, triggering an inflammatory response with the subsequent release of proinflammatory cytokines in various organs. In the arterial vessel wall, it leads to endothelial cell dysfunction, the oxidation of LDL, and the formation of atheromatous plaque. Additionally, HIV directly contributes to dyslipidemia by increasing the levels of FFA and VLDL, while also decreasing the functionality of HDL and impeding reverse cholesterol transport. Furthermore, the use of HAART exacerbates and intensifies dyslipidemia by inducing liver steatosis, promoting the accumulation of fatty acids, fostering lipogenesis, causing abnormalities in adipocyte metabolism, and leading to lipodystrophy. HIV, human immunodeficiency virus; HAART, highly active antiretroviral therapy; DAMP, damage-associated molecular pattern; PAMP, pathogen-associated molecular pattern; LPS, lipopolysaccharides; TG, triglyceride; VLDL, very low density lipoprotein; HLD, high-density lipoprotein; LDL, low-density lipoprotein; RCT, reverse cholesterol transport; NO, nitric oxide; FFA, free fatty acid; TNF-α, tumor-necrosis factor-α; PPARγ, peroxisome proliferator-activated receptor γ; MTP, microsomal triglyceride transfer protein; LXR, liver X receptors; IL, interleukin; MCP, monocyte chemoattractant protein; ATP, adenosine triphosphate; LPL, lipoprotein lipase; APO, apolipoprotein; TLR, toll-like receptor; ABCA1, adenosine triphosphate-binding cassette transporter A1.

**Table 1 life-14-00449-t001:** Summary of the effect of individual antiretroviral drugs on lipid parameters.

Drug Class	Antiretroviral Drug	Total Cholesterol	LDL-C	HDL-C	Triglycerides
Protease inhibitors (PIs)	Atazanavir/ritonavir	↔	↑	↔	↑
	Darunavir/ritonavir	↔	↑	↔	↑
	Indinavir	↑	↑	↑	↑
	Lopinavir/ritonavir	↑↑	↑	↔	↑↑
	Nelfinavir	↑	↑↑	↔	↑
Nucleotide reverse transcriptase inhibitors (NRTIs)	Abacavir	↑	↑	↔	↑
	Zidovudine	↑	↑	↔	↑
	Emtricitabine	↔	↔	↔	↔
	Lamivudine	↔	↔	↔	↔
	Stavudine	↑	↑	↓	↑
	Tenofovir alafenamide	↔	↑	↑	↑
	Tenofovir disoproxil	↓	↔	↓	↔
Non-nucleotide reverse transcriptase inhibitors (NNRTIs)	Efavirenz	↑	↑	↑	↑
	Etravirine	↔	↔	↔	↔
	Nevirapine	↑	↑	↑↑	↑
	Rilpivirine	↑	↑	↔	↔
Integrase strand transfer inhibitors (INSTIs)	Raltegravir	↔	↔	↑	↓
	Dolutegravir	↔	↔	↑	↓
	Bictegravir	↑	↓	↑	↓
	Cabotegravir	↓	↔	↑	↓

↑ = some increase; ↑↑ = moderate increase; ↓ = some decrease; ↔ = no significant change; LDL-C, low-density lipoprotein cholesterol; HDL-C, high-density lipoprotein cholesterol.

**Table 2 life-14-00449-t002:** Summary of major studies evaluating the treatment options for HIV-associated dyslipidemia.

Study/Year	(*n*)	Study Aim	Subject Characteristics	Clinical Outcome
Li et al. (2021) [147]	36,253	Effect of statins on the risk of CVD, cancer, and all-cause mortality	PLHIV under stable HAART	Statins reduced the risk of cancer and mortality but not CVD
Funderburg et al. (2014) [148]	147	Effect of rosuvastatin vs. placebo on inflammatory markers of CVD	PLHIV under stable HAART, HIV RNA < 1000 copies/mL, LDL ≤ 130 mg/dL, TG ≤ 500 mg/dL, without CVD/DM	Rosuvastatin decreased sCD14 and proportions of CD14^Dim^ and CD16+ monocytes
Gili et al. (2016) [12]	736	Efficacy and safety of statins on TC, LDL, HDL, and TG levels	PLHIV under stable HAART	Reduced TC with rosuvastatin and atorvastatin; reduced LDL with rosuvastatin, atorvastatin, and simvastatin; reduced TG with rosuvastatin, simvastatin, and atorvastatin; and increased HDL with pravastatin, rosuvastatin, and atorvastatin
Grinspoon et al. (2023) [149]	7769	Effect of pitavastatin vs. placebo on MACEs	PLHIV under stable HAART with low-to-moderate CVD risk	Incidence of MACEs was significantly lower in the pitavastatin group vs. the placebo group. Major outcome led to early termination of the study
Saeedi et al. (2015) [150]	43	Effect of ezetimibe/rosuvastatin vs. rosuvastatin on apoB, LDL, TC, TG, HDL, non-HDL, apoA1, apoB/apoA1, TC/HDL and CRP levels	PLHIV under stable HAART with apoB >80 mg/dL	ApoB, TC, TG, and non-HDL levels reduced more significantly in ezetimibe/rosuvastatin vs. rosuvastatin group
Boccara et al. (2022) [151]	467	Efficacy of evolocumab on the reduction of LDL, TG, non-HDL, apoB, TC, VLDL, and Lp(a) and potential increase in HDL levels	PLHIV under stable HAART with mean LDL of 133 mg/dL, CVD, DM, and intermediate/high 10-year-ASCVD risk	Significant decrease in LDL, TG, non-HDL, apoB, TC, VLDL, and Lp(a) levels, with concomitant increase in HDL levels
Nissen et al. (2023) [152]	13,970	Efficacy of bempedoic acid on MACEs (cardiovascular causes, nonfatal myocardial infarction, nonfatal stroke, and coronary revascularization) and LDL levels	Statin-intolerant with mean LDL of 139 mg/dL and high CVD risk	Bempedoic acid reduced LDL and incidence of all MACE points, except nonfatal stroke, death from CVD, and death from any cause
Muñoz et al. (2013) [153]	493	Efficacy of fish oil, fenofibrate, gemfibrozil, and atorvastatin on TG levels	PLHIV under stable HAART	All treatment options reduced TG levels, with fibrates being more effective and atorvastatin less effective than fish oils

PLHIV, people living with HIV; HAART, highly active antiretroviral treatment; LDL, low-density lipoprotein; TG, triglyceride; TC, total cholesterol; HDL, high-density lipoprotein; non-HDL, non-high-density lipoprotein; VLDL, very low density lipoprotein; apoB, apolipoprotein B; apoA1, apolipoprotein A1; Lp(a), lipoprotein a; CRP, C-reactive protein; sCD14, soluble CD14; sCD163, soluble CD163; CVD, cardiovascular disease; ASCVD, atherosclerotic cardiovascular disease; DM, diabetes mellitus; MACE, major adverse cardiac event; ACC/AHA, American College of Cardiology/American Heart Association.

**Table 3 life-14-00449-t003:** The effect of lipid-lowering agents on the serum concentration of antiretroviral drugs.

Hypolipidemic Agent	PIs	NNRTIs	NRTIs	INSTIs	Recommendations
Rosuvastatin	↑ C with ATV/r, LPV/r, DRV/r	-	-	↑ C with EVG/c	Initial dose 5 mg with slow titration, do not exceed 20 mg with cobicistat-boosted drugs
Fluvastatin	↓ C with NFV	↓ C with EFV	-	-	Initial dose > 20 mg
Pravastatin	↓ C with SQV/r, ↑ C with LPV/r, DRV/r	↓ C with EFV	-	-	Suitable dose adjustment
Lovastatin	Contraindicated	Possible ↑ C	-	Contraindicated with EVG/c	Consider low initial dose
Simvastatin	Contraindicated	Possible ↑ C	-	Contraindicated with EVG/c	Consider low initial dose
Atorvastatin	↑ C with SQV/r, LPV/r	↓ C with EFV, ETV	-	↑ C with EVG/c	Dose 10–40 mg for PIs, 40–80 mg for NNRTIs, lowest effective dose for cobicistat-boosted drugs
Pitavastatin	-	-	-	-	No dose adjustment
Ezetimibe	-	-	-	-	No dose adjustment
PCSK9i	-	-	-	-	No dose adjustment
Bempedoic acid	-	-	-	-	No dose adjustment
Fenofibrate	-	-	-	-	No dose adjustment
Gemfibrozil	↑ C with LPV/r	-	-	-	Consider low initial dose
Fish oils	-	-	-	-	Possible pill burden

↑, increase; ↓, decrease; PIs, protease inhibitors; NNRTIs, non-nucleoside reverse transcriptase inhibitors; NRTIs, nucleoside reverse transcriptase inhibitors; INSTIs, integrase inhibitors; C, concentration; ATV/r, atazanavir/ritonavir; LPV/r, lopinavir/ritonavir; DRV/r, darunavir/ritonavir; EVG/c, elvitegravir/cobicistat; NFV, nelfinavir; EFV, efavirenz; SQV/r, saquinavir/ritonavir.

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
