# Peer review of "Pathophysiology and Clinical Management of Dyslipidemia in People Living with HIV: Sailing through Rough Seas"

_life, 2024, doi:10.3390/life14040449_

Round 1

Reviewer 1 Report

Comments and Suggestions for Authors

This is a well written narrative review focusing on the mechanisms and therapeutic algorithms in dyslipidemia among HIV infected patients. I only have some minor suggestions so as to increase the visibility of this manuscript.

1) I would change the title in section 2.1 The role of HIV viremia to something broader since in this section the role of inflammation and gut microbiota are mentioned. Alternatively, you can create smaller sub-sections. 

2) You can also include a table summarizing the major drug-drugs interactions

3) Provide a table summarizing the major trials and meta-analyses regarding the safety and efficacy of the lipid-lowering therapies (LLT) among HIV patients. You have already mentioned them throughout your manuscript, but I think that a table including the trials that examing a specific LLT (statins, bempedoic, fibates etc)  among HIV patients is highly needed (provide number of HIV patients, major comorbidities and cardiovascular factors and clinical outcomes)

Author Response

Please also see the attachment.

Reviewer #1: This is a well written narrative review focusing on the mechanisms and therapeutic algorithms in dyslipidemia among HIV infected patients. I only have some minor suggestions so as to increase the visibility of this manuscript.

Point 1. I would change the title in section 2.1 The role of HIV viremia to something broader since in this section the role of inflammation and gut microbiota are mentioned. Alternatively, you can create smaller sub-sections.

Response: We thank the reviewer for the suggestion. Section 2.1 has been renamed to “2.1. The Role of HIV Viremia and Inflammation”

Point 2. You can also include a table summarizing the major drug-drugs interactions

Response: We thank the reviewer for the comment. According to the suggestion, we have added Table 3. entitled: “The effect of lipid-lowering agents on the serum concentration of antiretroviral drugs, which summarizes the drug-drug interactions”.

Point 3. Provide a table summarizing the major trials and meta-analyses regarding the safety and efficacy of the lipid-lowering therapies (LLT) among HIV patients. You have already mentioned them throughout your manuscript, but I think that a table including the trials that examing a specific LLT (statins, bempedoic, fibates etc) among HIV patients is highly needed (provide number of HIV patients, major comorbidities and cardiovascular factors and clinical outcomes)

Response: We thank the reviewer for the suggestion. We have added Table 2. entitled “Summary of major studies evaluating the treatment options for HIV-associated dyslipidemia”, which summarizes the major trials and meta-analyses regarding the safety and efficacy of the lipid-lowering therapies (LLT) among HIV patients.

Reviewer 2 Report

Comments and Suggestions for Authors

Papantoniou et al. present a timely and well written review on dyslipidemia and HIV. Overall, the review integrates a significant amount of published work and is very informative and relevant. Great Figure and Table. Overall, I have no major concerns and the author’s should be commended on the work. Perhaps consider shortening the title a little bit and expanding the conclusions to include a recommendation or best-practices statement regarding overall treatment? Other minor comments are below.

Minor:

1.     In the abstract, line 24, what is meant by “induced acquired immune deficiency”.

2.     In the abstract, line 28 – “Among the HAART regimens…..”. This sentence is too long and very hard to understand. It needs to be rewritten and/or broken up.  

3.     Line 111 – TNFa needs the “a” italicized.

4.     Line 118 – what is meant by “cholesterol-pumped macrophages”?

5.     Line 122 – what is TNF-Z?

6.     Line 135 – “IFN-a and TNF-a” needs the “a” italicized.

7.     Line 200 – what is meant by “In the cross-sectional D: A: D”

8.     Line 262 – “mg bid instead of 40 mg b.i.d.” needs the first bid with periods.

9.     Line 564 – 3.3 title has Bempedoic Acid spelled incorrectly.

-  - Consider the following articles as having potential relevance to article: PMID: 34681637, PMID: 33003981, and PMID: 35471692, especially with the dynamic interplay between HIV, obesity, and aging.

Author Response

Please also see the attachment.

Reviewer #2: Papantoniou et al. present a timely and well written review on dyslipidemia and HIV. Overall, the review integrates a significant amount of published work and is very informative and relevant. Great Figure and Table. Overall, I have no major concerns and the authors should be commended on the work. Perhaps consider shortening the title a little bit and expanding the conclusions to include a recommendation or best-practices statement regarding overall treatment? Other minor comments are below.

Response: We thank the reviewer for the positive and constructive feedback. According to the reviewer’s suggestions, we have shortened the title to “Pathophysiology and Clinical Management of Dyslipidemia in People Living with HIV: Sailing through Rough Seas”, and expanded the conclusions in order to include a recommendation and best-practices statement regarding overall treatment.

Point 1. In the abstract, line 24, what is meant by “induced acquired immune deficiency”.

Response: We thank the reviewer for the comment. The word “induced” was deleted.

Point 2. In the abstract, line 28 – “Among the HAART regimens….”. This sentence is too long and very hard to understand. It needs to be rewritten and/or broken up. 

Response: We thank the reviewer for the suggestion. The corresponding sentence has been rephrashed to “Amongst HAART regimens, darunavir and atazanavir, tenofovir disoproxil fumarate, nevirapine, rilpivirine, and especially integrase inhibitors have demonstrated the most favorable lipid profile, emerging as sustainable options in HAART substitution” in order to be easier to understand and shorter.

Point 3. Line 111 – TNFa needs the “a” italicized.

Response: We thank the reviewer for the correction, the text has been formatted accordingly.

Point 4. Line 118 – what is meant by “cholesterol-pumped macrophages”?

Response: We thank the reviewer for the query. The text has been rephrased to “cholesterol-overloaded dysfunctional macrophages”.

Point 5. Line 122 – what is TNF-Z?

Response: We thank the reviewer for the correction. The text has been corrected to TNF-a.

Point 6. Line 135 – “IFN-a and TNF-a” needs the “a” italicized.

Response: We thank the reviewer for the suggestion. The text has been reformated accordingly.

Point 7. Line 200 – what is meant by “In the cross-sectional D: A: D”

Response: We thank the reviewer for the correction, the term “D: A: D” has been deleted from the text.

Point 8. Line 262 – “mg bid instead of 40 mg b.i.d.” needs the first bid with periods.

Response: We thank the reviewer for the correction, the text has been modified accordingly.

Point 9. Line 564 – 3.3 title has Bempedoic Acid spelled incorrectly.

Response: We thank the reviewer for the correction, the text has been modified accordingly.

Point 10. Consider the following articles as having potential relevance to article: PMID: 34681637, PMID: 33003981, and PMID: 35471692, especially with the dynamic interplay between HIV, obesity, and aging.

Response: We thank the reviewer for the suggestion. All of the relevant articles have been cited at the revised version of the manuscript (References n. 4, 79, 80).

Reviewer 3 Report

Comments and Suggestions for Authors

The initiation of well-tolerated highly active antiretroviral therapy (HAART) has dramatically increased the survival for HIV-infected individuals. However, complications such as coronary heart disease associated with lipid toxicities has become an important comorbid condition especially in elderly living HIV subjects. Hence, effective management of dyslipidemia in this particular population is essential to reduce the burden of cardiovascular complications despite many challenges due to interactions between HAART and lipid-lowering medications.

The authors aimed to provide a narrative updated review on clinical evaluation and management of dyslipidemia in living people with HIV.

The review is well written and the authors have addressed some of the future aspects for effective management of dyslipidemia and cardiovascular disease risks associated with HIV. They also proposed future directions focusing on better understanding of molecular mechanisms involved in HIV-associated dyslipidemia.

Author Response

Please also see the attachment.

Reviewer #3: The initiation of well-tolerated highly active antiretroviral therapy (HAART) has dramatically increased the survival for HIV-infected individuals. However, complications such as coronary heart disease associated with lipid toxicities has become an important comorbid condition especially in elderly living HIV subjects. Hence, effective management of dyslipidemia in this particular population is essential to reduce the burden of cardiovascular complications despite many challenges due to interactions between HAART and lipid-lowering medications.

The authors aimed to provide a narrative updated review on clinical evaluation and management of dyslipidemia in living people with HIV.

The review is well written and the authors have addressed some of the future aspects for effective management of dyslipidemia and cardiovascular disease risks associated with HIV. They also proposed future directions focusing on better understanding of molecular mechanisms involved in HIV-associated dyslipidemia.

Response: We thank the reviewer for the positive and constructive feedback.
